# ADIS-GAN: AFFINE DISENTANGLED GAN

## ABSTRACT

This paper proposes Affine Disentangled GAN (ADIS-GAN), which is a Generative Adversarial Network that can explicitly disentangle affine transformations in a self-supervised and rigorous manner. The objective is inspired by InfoGAN, where an additional affine regularizer acts as the inductive bias. The affine regularizer is rooted in the affine transformation properties of images, changing some properties of the underlying images, while leaving all other properties invariant. We derive the affine regularizer by decomposing the affine matrix into separate transformation matrices and inferring the transformation parameters by maximum likelihood estimation. Unlike the disentangled representations learned by existing approaches, the features learned by ADIS-GAN are axis-aligned and scalable, where transformations such as rotation, horizontal and vertical zoom, horizontal and vertical skew, horizontal and vertical translation can be explicitly selected and learned. ADIS-GAN successfully disentangles these features on the MNIST, CelebA, and dSprites datasets.

## 1 INTRODUCTION

In a disentangled representation, observations are interpreted in terms of a few explanatory factors. Examples of such factors are the rotation angle(s), scale, or position of an object in an image. Disentangled variables are generally considered as the abstraction of interpretable semantic information and reflection of separatable factors of variation in the data. Many studies have explored the effectiveness of disentangled representations (Bengio et al., 2013; N et al., 2017; LeCun et al., 2015; Lake et al., 2017; Tschannen et al., 2018). The information presented in observations is encoded in an interpretable and compact manner, e.g., the texture style and the orientation of the objects (Bengio et al., 2013; LeCun et al., 2015; Lake et al., 2017; Tschannen et al., 2018). The learned representation are more generalizable and can be useful for downstream tasks, such as classification and visualization (Bengio et al., 2013; N et al., 2017; Chen et al., 2016).

The concept of disentangled representation has been defined in several ways in the literature (Locatello et al., 2019; Higgins et al., 2018; Eastwood & Williams, 2018). The necessity of explicit inductive biases both for learning approaches and the datasets is discussed in Locatello et al. (2019). Inductive bias refers to the set of assumptions that the learner uses to predict outputs of given inputs that it has not encountered. For instance, in the dSprites dataset objects are displayed at different angles and positions; such prior knowledge helps to detect and classify the objects. However, the inductive biases in existing deep learning models are mostly implicit. The proposed ADIS-GAN utilizes relative affine transformations (see Section 4) as the explicit inductive bias, leading to axis-aligned and scalable disentangled representations.

**Axis-alignment:** The issue of axis-alignment is addressed in Higgins et al. (2018), where each latent dimension should have a pre-defined unique axis-alignment. Without axis-alignment, the features learned by disentangled representation need to be identified with expert knowledge after the training, which could be a cumbersome process when dealing with a large number of features. The axis-alignment property also helps to discover desired but non-dominant attributes (e.g., roll angle of face in CelebA dataset).

**Scalability:** The scalability property allows us to make a trade-off between the compactness and expressivity of the disentangled representation. For example, the zoom attribute can be decomposed as horizontal and vertical zoom. A more compact representation encodes the zoom attribute by one latent dimension, while a more expressive representation decomposes the zoom attribute as

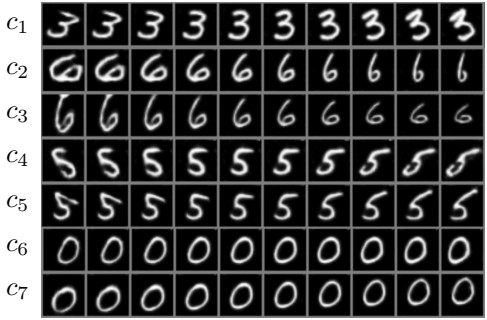

Figure 1: Disentangled representation of affine transformations on the MNIST dataset. $c_1$: rotation, $c_2$: horizontal zoom, $c_3$: vertical zoom, $c_4$: horizontal skew, $c_5$: vertical skew, $c_6$: horizontal translation, $c_7$: vertical translation. To the best of our knowledge, ADIS-GAN is the first algorithm that can disentangle an entire affine transformation in a self-supervised manner.

horizontal and vertical zoom, encoded by two latent dimensions. The scalability property provides a solution to an open question related to disentangled representations: Should we capture a data generative factor (e.g., zoom) by one or multiple latent dimensions? Many disentanglement metrics Higgins et al. (2017); Chen et al. (2018); Kim & Mnih (2018); Eastwood & Williams (2018) rely on a single latent dimension, while others Denton & Birodkar (2017); Ridgeway & Mozer (2018); Higgins et al. (2018) encode latent factors via multiple latent dimensions.

We motivate the importance of axis-alignment and scalability in particular for affine transformations (see Figure 1), where disentangling object poses from texture and shape is an attractive property of an algorithm in the imaging domain (Jaderberg et al., 2015; Bepler et al., 2019; Engstrom et al., 2019). In supervised learning tasks, *Spatial Transformer Network* Jaderberg et al. (2015) can actively spatially transform an image by providing a proper affine transformation matrix. In unsupervised learning tasks, few algorithms have successfully disentangled the affine transformation. In Bepler et al. (2019), an algorithm is introduced that disentangles rotation and translation but not an entire affine transformation.

We propose ADIS-GAN, which is a Generative Adversarial Network that utilizes the affine regularizer (see Section 4) as an inductive bias to explicitly disentangle the affine transformation. The affine regularizer is rooted in the affine transformation properties of the images, that affect certain properties of the underlying images, while leaving all other properties invariant. We derive the affine regularizer by decomposing the affine matrix into separate transformations and inferring the transformation parameters by maximum likelihood estimation. Unlike the disentangled representations learned by existing approaches, the features learned by ADIS-GAN are axis-aligned and scalable, where transformations such as rotation, horizontal and vertical zoom, horizontal and vertical skew, horizontal and vertical translation can be explicitly selected and learned (see Figure 1).

In the remainder of the paper, we review related work in Section 2. We then compare the difference between GAN, InfoGAN and the proposed method in Section 3. We introduce the ADIS-GAN in Section 4, while in Section 5, we show numerical results showing the axis-aligned and scalable disentangled representation learned by ADIS-GAN. We offer concluding remarks in Section 6.

**Our contributions:**

1. To the best of our knowledge, ADIS-GAN is the first algorithm that can disentangle an entire affine transformation, including rotation, horizontal and vertical zoom, horizontal and vertical skew, horizontal and vertical translation in a self-supervised manner.

2. The disentangled representations obtained by ADIS-GAN are axis-aligned. The advantages are two-folds: a. The attributes to be learned can be pre-defined, which saves the effort to identify the attributes after the training. b. Desired but non-dominant attributes can be learned, in parallel with the dominant attributes.

3. The disentangled representations obtained by ADIS-GAN are scalable. The scalability property makes it possible to make a trade-off between the compactness and expressivity of the learned representation.

## 2 RELATED LITERATURE

Recent approaches learning disentangled representations are largely based on *Variational Autoencoders (VAEs)* Kingma & Welling (2013) and *InfoGAN* Chen et al. (2016). In a standard *VAE*, observation $x$ is encoded to latent representation $z$ from variational distribution $Q(z|x)$ using a deep neural network. In the generative process, the latent variable $z$ is sampled from a prior distribution $P(z)$ and uses another deep neural network to reconstruct the observation $x$ from conditional distribution $P(x|z)$. To achieve the disentanglement, a factorized aggregated posterior $\int_x Q(z|x)P(x)dx$ is encouraged during the training. In a standard *GAN* Goodfellow et al. (2014), latent representation $z$ is sampled from a prior distribution $P(z)$. The fake data $x_{\text{fake}}$ is generated by $z$ from distribution $P(x|z)$. A discriminator is introduced to encourage the generated data $x_{\text{fake}}$ to be close to the observation $x_{\text{real}}$. To achieve disentanglement, *InfoGAN* Chen et al. (2016) proposes to maximize the mutual information between a subset $c$ of latent representation $z$ and the generated data $x_{\text{fake}}$.

Recently, much attention has been paid to regularizers that promote disentanglement. The $\beta$-VAE Higgins et al. (2017) encourages the disentanglement by increasing the weight of the KL regularizer, thus promoting the factorization of the posterior $Q(z|x)$. Both FactorVAE Kim & Mnih (2018) and $\beta$-TCVAE Chen et al. (2018) penalize the total correlation, while the former relies on adversarial training and the latter directly calculates the total correlation through the decomposition of the $\beta$-VAE objective function. The HFVAE Esmaeili et al. (2019) proposes a two-level hierarchical objective to control relative degree of statistical independence. In the ChyVAE Ansari & Soh (2019), an inverse-Wishart (IW) prior on the covariance matrix of the latent code is augmented to promote statistical independence. The DIP-VAE Kumar et al. (2018) penalizes the difference between the aggregated posterior and a factorized prior. In the AnnealedVAE Huang et al. (2018), the encoder can concentrate on learning individual factors and variation by gradually increasing the bottleneck capacity. The IB-GAN Jeon et al. (2019) is an extension to InfoGAN rooted in information bottleneck theory, which includes a mutual information upper bound and forms a mutual information bottleneck. The InfoGAN-CR Lin et al. (2019) adds a contrastive regularizer on top on InfoGAN, that compares the changes between the image and latent space.

Even though the aforementioned methods have achieved better disentanglement performance compared to the baseline established by VAE and InfoGAN, none of them have successfully disentangled an entire affine transformations in a scalable and axis-aligned way, which is a desirable property in the imaging domain. Moreover, our affine regularizer is orthogonal to those approaches, thus makes it possible to integrate our affine regularizer with other methods.

In Gidaris et al. (2018); Chen et al. (2019); Wang et al. (2020); Zhang et al. (2019), self-supervised regularization is applied, where they compare the difference of images before and after the affine/projective transformation. The transformation loss is define as: $L = ||M(\theta') - M(\theta)||_2^2$ and a parameterized matrix $M \in \mathbb{R}^{3 \times 3}$. We make a more specific assumption about the matrix and further decompose it to achieve disentanglement (see Section 4).

## 3 BACKGROUND: GAN, INFOGAN AND ADIS-GAN

In a standard *GAN* Goodfellow et al. (2014), latent representation $z$ is sampled from a prior distribution $P(z)$. The fake data $x_{\text{fake}}$ is generated from $z$ from conditional distribution $P(x|z)$. A discriminator is introduced to encourage the generated data $x_{\text{fake}}$ to be close to the observation $x_{\text{real}}$:

$$\min_G \max_D V(D, G) = L_{\text{adv}}(D, G). \tag{1}$$

To achieve disentanglement, *InfoGAN* Chen et al. (2016) maximizes the mutual information between a subset $c$ of latent representation $z$ and the generated data $x_{\text{fake}}$:

$$\min_G \max_D V(D, G) = L_{\text{adv}}(D, G) - \lambda I(c'; x_{\text{fake}}). \tag{2}$$

To explicitly disentangle affine transformation attributes in a scalable and axis-aligned manner, ADIS-GAN has one additional term called affine regularizer on top of InfoGAN:

$$\min_G \max_D V(D, G) = L_{\text{adv}}(D, G) - \lambda I(c'; x_{\text{fake}}) - \beta L_{\text{affine}}. \tag{3}$$

## 4 AFFINE REGULARIZER

Compared to InfoGAN, the major difference of ADIS-GAN is the affine regularization loss $L_{\text{affine}}$ (see Figure 2). To calculate $L_{\text{affine}}$, three modifications to the network have been made. 1: Besides generating $x_{\text{fake}}$ and calculating mutual information loss $I(c'; x_{\text{fake}})$ like InfoGAN, the semantic latent vector $c$ of ADIS-GAN has an additional task: to generate random affine transformation parameters. We now rename the latent vector $c$ as $c_{\text{transform}}^{\text{real}}$ to describe the state before and after the affine transformation, where superscript "real" stands for the initial state before the transformation and subscript "transform" stands for the final state after the transformation, more details regarding the transformation states will be introduced in Section 4.2. $c_{\text{transform}}^{\text{real}}$ is sampled from a uniform distribution, then converted to an affine matrix $M_{\text{transform}}^{\text{real}}$ using Flow 1 (see Figure 4). 2: Affine transformation augmented image $x_{\text{transform}}$ is introduced. $x_{\text{transform}}$ is obtained by multiplying $M_{\text{transform}}^{\text{real}}$ and $x_{\text{real}}$ sampled from training data (see Figure 3). Unlike InfoGAN where "$x_{\text{real}}$" is the positive sample fed to the discriminator. In ADIS-GAN,"$x_{\text{transform}}$" is the positive sample fed to the discriminator. This guarantees the affine transformations are observed by the network. 3: Affine reugularization loss $L_{\text{affine}}$ is calculated by comparing the ground truth affine transformation latent vector $c_{\text{transform}}^{\text{real}}$ and the reconstructed affine transformation latent vector $c_{\text{transform}}^{\text{real}'}$, which is obtained by calculating the relative difference between $x_{\text{real}}$ and its affine augmented pair $x_{\text{transform}}$ (see Figure 3).

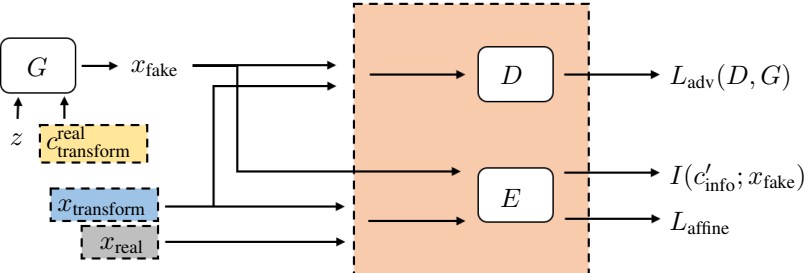

Figure 2: The main framework of ADIS-GAN. D stands for discriminator, E stands for encoder, G stands for generator. $x_{\text{fake}}$ is the generated image, $x_{\text{real}}$ is the image sampled from training dataset, $x_{\text{transform}}$ is the affine transformed image. The generation process of $x_{\text{transform}}$ is demonstrated in Figure 3. $z$ is latent noise sampled from normal distribution. $c_{\text{transform}}^{\text{real}}$ is the semantic latent vectors sampled from uniform distribution. $c_{\text{transform}}^{\text{real}}$ has two usages: 1. It is used to generate $x_{\text{fake}}$ and compute mutual information loss. 2. It is used to generate $x_{\text{transform}}$ together with $x_{\text{real}}$ and compute affine regularization loss (see Figure 3). The meaning of the superscript "real" refers to the initial state before the transformation and subscript "transform" refers to the final state after the transformation. $I(c'_{\text{info}}; x_{\text{fake}})$ is the mutual informaion loss. $L_{\text{adv}}(D, G)$ is the GAN loss. $L_{\text{affine}}$ is the affine regularization loss. Figure 3 illustrates how to calculate the affine regularization loss $L_{\text{affine}}$.

The working principle of affine regularizer is illustrated in Figure 3. First, the latent vector $c_{\text{transform}}^{\text{real}}$ is sampled from a uniform distribution. It is converted to an affine matrix $M_{\text{transform}}^{\text{real}}$ through "Flow 1" (see Section 4.1.1). Next, the real image "$x_{\text{real}}$" sampled from training data is affine transformed with $M_{\text{transform}}^{\text{real}}$ to obtain $x_{\text{transform}}$. Both $x_{\text{real}}$ and its transformed pair $x_{\text{transform}}$ are encoded to latent vectors $c_{\text{real}}^{\text{basis}}$ and $c_{\text{transform}}^{\text{basis}}$ respectively. The encoded latent vectors $c_{\text{real}}^{\text{basis}}$ and $c_{\text{transform}}^{\text{basis}}$ are further converted to affine matrix $M_{\text{real}}^{\text{basis}}$ and $M_{\text{transform}}^{\text{basis}}$ through "Flow 1" (see Section 4.1.1). Then the relative transformation matrix $M_{\text{transform}}^{\text{real}'}$ is computed using both $M_{\text{real}}^{\text{basis}}$ and $M_{\text{transform}}^{\text{basis}}$ (see Section 4.2 Equation 9). Finally, the reconstructed latent vector $c_{\text{transform}}^{\text{real}'}$ is obtained using $M_{\text{transform}}^{\text{real}'}$ through "Flow 2" (see Section 4.1.2). Here we use rotation ($\theta$), horizontal and vertical zoom ($p, q$), horizontal and vertical translation ($x, y$) as an example (more affine parameter combinations can be found in Appendix H). The affine regularizer loss is $L_{\text{affine}} = \sum_i ||c(i)_{\text{transform}}^{\text{real}} - c(i)_{\text{transform}}^{\text{real}'}||_2^2, i \in \{\theta, p, q, x, y\}$. You may refer to Algorithm 1 and Figure 3 for more details.

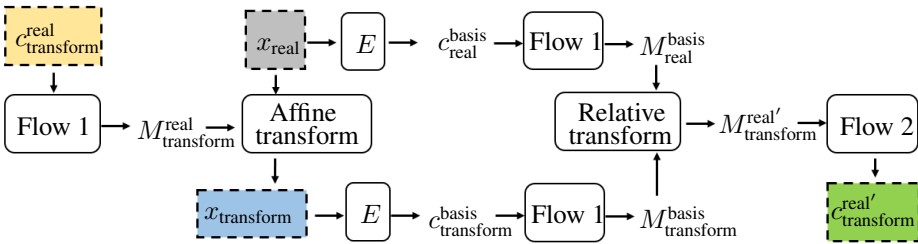

Figure 3: Affine regularizer diagram. Inputs: $c_{\text{transform}}^{\text{real}}$ sampled from Unif(-1,1) and $x_{\text{real}}$ sampled from training data. Output: $x_{\text{transform}}$ and $c_{\text{transform}}^{\text{real}'}$. The affine regularizer loss is: $L_{\text{affine}} = \sum_i ||c(i)_{\text{transform}}^{\text{real}} - c(i)_{\text{transform}}^{\text{real}'}||_2^2, i \in \{\theta, p, q, x, y\}$. $\theta$: rotation, $p$: horizontal zoom, $q$: vertical zoom, $x$: horizontal translation, $y$: vertical translation. E stands for encoder, Flow 1 and Flow 2 are defined in Section 4.1 and Figure 4. Relative transform is defined in Section 4.2 Equation 9.

---

**Algorithm 1:** Affine Regularizer

**Input:** Sampled images: $x_{\text{real}}$, latent vectors: $c(i)_{\text{transform}}^{\text{real}} \sim$ Unif(-1,1), $i \in \{\theta, p, q, x, y\}$
**Output:** $L_{\text{affine}}$
**Matrix from sampled latent vector:** $M_{\text{transform}}^{\text{real}} = \text{Flow } 1(c(i)_{\text{transform}}^{\text{real}})$, $i \in \{\theta, p, q, x, y\}$
**Affine transformation:** $x_{\text{transform}} = M_{\text{transform}}^{\text{real}} \times x_{\text{real}}$
**Latent vectors encoded from images:** $c_{\text{transform}}^{\text{basis}} = \text{Encoder}(x_{\text{transform}})$, $c_{\text{real}}^{\text{basis}} = \text{Encoder}(x_{\text{real}})$
**Matrices from encoded latent vector:** $M_{\text{transform}}^{\text{basis}} = \text{Flow } 1(c(i)_{\text{transform}}^{\text{basis}})$, $M_{\text{real}}^{\text{basis}} =$
  Flow $1(c(i)_{\text{real}}^{\text{basis}})$, $i \in \{\theta, p, q, x, y\}$
**Relative transform matrix:** $M_{\text{transform}}^{\text{real}'} = M_{\text{transform}}^{\text{basis}} \times M_{\text{real}}^{\text{basis}^{-1}}$
**Reconstructed latent vectors:** $c(i)_{\text{transform}}^{\text{real}'} = \text{Flow } 2(M_{\text{transform}}^{\text{real}'})$
**Affine reuglarizer loss:** $L_{\text{affine}} = \sum_i ||c(i)_{\text{transform}}^{\text{real}} - c(i)_{\text{transform}}^{\text{real}'}||_2^2, i \in \{\theta, p, q, x, y\}$

---

Figure 4: Conversion between latent vectors and affine matrix. Flow 1: convert the latent vectors $c$ to affine matrix $M$. Flow 2: convert the affine matrix $M$ to latent vectors $c$. $c$ stands for semantic latent vectors. $\tilde{c}$ stands for affine parameters such as rotation ($\theta$), zoom ($p, q$) and translation ($x, y$). $M$ stands for affine matrix. "Norm$_{\text{LA}}$" and "M Initialization" are described in Section 4.1.1. "Norm$_{\text{AL}}$" and "MLE" are described in Section 4.1.2.

## 4.1 CONVERSION BETWEEN LATENT VECTORS AND AFFINE MATRIX

### 4.1.1 LATENT VECTORS TO AFFINE MATRIX: FLOW 1

To conduct affine transformation and calculate the relative difference, we need to convert latent vector $c$ to an affine matrix $M$. The "Flow 1" in Figure 4 illustrates the process.

**Norm$_{\text{LA}}$** (Latent to Affine): First, we need to normalize the latent vector $c$ to the affine transformation parameter $\tilde{c}$. For instance, if we set the affine transformation range as rotation $\theta \in [-\pi/10, \pi/10]$, horizontal and vertical zoom $p, q \in [0.8, 1.2]$, horizontal and vertical translation $x, y \in [-0.2, 0.2]$, and a 5-d latent vector $c$ is sampled from uniform distribution Unif$(-1, 1)$. **Norm$_{\text{LA}}$** (Latent to Affine) refers to the following normalization: $\tilde{c}(\theta) = c(\theta) \times (\pi/10)$, $\tilde{c}(p) = c(p) \times 0.2 + 1$, $\tilde{c}(q) = c(q) \times 0.2 + 1$, $\tilde{c}(x) = c(x) \times 0.2$, $\tilde{c}(y) = c(y) \times 0.2$.

**M Initialization**: Next, the affine parameter $\tilde{c}$ needs to be properly arranged in the affine matrix $M$. By default, the affine matrix $M$ is organized as in Equation 4.

$$M = \begin{bmatrix} A_{11} & A_{12} & A_{13} \\ A_{21} & A_{22} & A_{23} \\ 0 & 0 & 1 \end{bmatrix} = \begin{bmatrix} \cos\theta & -\sin\theta & 0 \\ \sin\theta & \cos\theta & 0 \\ 0 & 0 & 1 \end{bmatrix} \begin{bmatrix} p & 0 & 0 \\ 0 & q & 0 \\ 0 & 0 & 1 \end{bmatrix} \begin{bmatrix} 1 & 0 & x \\ 0 & 1 & y \\ 0 & 0 & 1 \end{bmatrix}. \tag{4}$$

### 4.1.2    AFFINE MATRIX TO LATENT VECTORS: FLOW 2

To obtain the reconstructed affine transformation latent vector "$c^{real'}_{transform}$", we need to convert the affine matrix to latent vector. The "Flow 2" in Figure 4 illustrates the process.

**MLE (Maximum Likelihood Estimation)**: First, we need to calculate the affine parameter $\tilde{c}$ using maximum likelihood estimation (see Appendix A).

$$\tilde{c}(i) = \begin{cases} \theta & = \arctan\frac{2(A_{11}A_{21}-A_{12}A_{22})}{A_{11}^2+A_{22}^2-A_{12}^2-A_{21}^2}, \\ p & = A_{11}\cos\theta + A_{21}\sin\theta, \\ q & = -A_{12}\sin\theta + A_{21}\sin\theta, \\ x & = \frac{A_{13}\cos\theta+A_{23}\sin\theta}{p}, \\ y & = \frac{-A_{13}\sin\theta+A_{23}\cos\theta}{q}, \end{cases} \tag{5}$$

where $i \in \theta, p, q, x, y$.

**Norm$_{AL}$ (Affine to Latent)**: Next, we need to normalize the affine parameter $\tilde{c}$ to latent vector $c$. If we apply the same affine transformation setting as mentioned in Section 4.1.1 Flow 1. **Norm$_{AL}$** (Affine to Latent) refers to the following normalization: $c(\theta) = \tilde{c}(\theta)\times(10/\pi)$, $c(p) = (\tilde{c}(p)-1)\times 5$, $c(q) = (\tilde{c}(q)-1)\times 5$, $c(x) = \tilde{c}(x)\times 5$, $c(y) = \tilde{c}(y)\times 5$.

## 4.2    RELATIVE AFFINE TRANSFORMATION

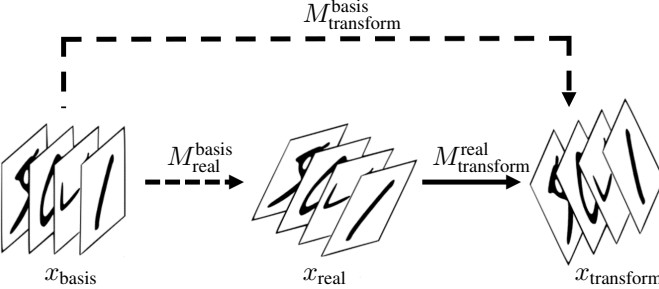

Figure 5: Illustration of relative affine transformation. Solid line stands for the affine transformation between two real images. Dashed line stands for the affine transformation between one real image and the imaginary image basis.

We assume, without loss of generality, that images can be expressed as the multiplication of the affine transformation $M^{basis}_{real} \in \mathbb{R}^{3\times3}$ (affine matrix is a $3 \times 3$ matrix by default) and an image basis $x_{basis}$ (see Figure 5):

$$M^{basis}_{real} \times x_{basis} = x_{real}. \tag{6}$$

The superscript "basis" of $M^{basis}_{real}$ stands for the initial state and subscript "real" stands for the transformed state of the image $x$. The image basis $x_{basis}$ is the average manifold of all images within the same category. For instance, the digits "0" , "1", ... "9" in MNIST are different categories. $x_{basis}$ does not refer to a particular image in the dataset and it is purely learned from the data.

If we purposely apply a known affine transformation $M_{\text{transform}}^{\text{real}}$ on a sampled image $x_{\text{real}}$, we can obtain an new image $x_{\text{transform}}$:

$$M_{\text{transform}}^{\text{real}} \times x_{\text{real}} = x_{\text{transform}}. \tag{7}$$

According to the definition in equation 6, the transformed image can also be expressed as the multiplication of the $M_{\text{transform}}^{\text{basis}}$ and the image base $x_{\text{basis}}$:

$$M_{\text{transform}}^{\text{basis}} \times x_{\text{basis}} = x_{\text{transform}}. \tag{8}$$

By solving the simultaneous formula from equation 6 to 8:

$$M_{\text{transform}}^{\text{real}} = M_{\text{transform}}^{\text{basis}} \times M_{\text{real}}^{\text{basis}\,-1}. \tag{9}$$

$M_{\text{transform}}^{\text{real}}$ is the **relative affine transformation** between affine matrices $M_{\text{transform}}^{\text{basis}}$ and $M_{\text{real}}^{\text{basis}}$.

## 5 NUMERICAL RESULTS

The goal of the experiments in this section is to investigate, both qualitatively and quantitatively, the disentangled representations obtained by ADIS-GAN. The axis-alignment of the disentangled representations is demonstrated on CelebA dataset Liu et al. (2015), while the scalability of the disentangled representation is illustrated on MNIST dataset. Next quantitative results for ADIS-GAN are presented and compared to benchmarks on the dSprites Matthey et al. (2017) dataset. The parameters of the affine transformation are selected as follows: rotation range: $[-18°, 18°]$, zoom range: [0.8, 1.2], and translation range: [-0.2, 0.2]. For all the experiments we use Adam optimizer with the learning rate of 0.0002 for discriminator and 0.0001 for generator and encoder. The batch size is 128 for MNIST, 64 for dSprites, and 16 for CelebA. The regularization weight in Equation 3 is 1 for both $\lambda$ and $\beta$. You may refer to Appendix G for network structure details.

### 5.1 QUALITATIVE RESULTS

The axis-alignment and scalability properties are achieved by selecting different matrix initializations and their corresponding maximum likelihood estimation. You may refer to Appendix I for various ways of matrix initializations and their maximum likelihood estimation.

#### 5.1.1 AXIS-ALIGNMENT

In CelebA dataset, typical dominant attributes are azimuth, sunglasses, emotion, etc. Existing methods Chen et al. (2016); Kim & Mnih (2018); Chen et al. (2018) have successfully disentangled those attributes (see Appendix E). However, non-dominant attributes such as the roll, width and length of the face and relative position of face in the frame are rarely tackled. Thanks to the axis-aligned property, ADIS-GAN can explicitly learn those non-dominant attributes (see Figure 6, 7 and 8). It is interesting to note that ADIS-GAN does not rigidly perform affine transformations on the given images but understands the relationship between different facial components. For example, when we try to shorten the length of the face (vertical zoom and translation, see left images on both Figure 7 and 8), ADIS-GAN knows how to complete the bottom of the images with a neck to make the images look realistic. To disentangle facial attributes on the CelebA data, we choose to model the latent space with one categorical code, $c_{\text{cat}} \sim \text{Cat}(K = 10, p = 0.1)$ and 5 continuous codes $c_{\text{cont}} \sim \text{Unif(-1,1)}$.

#### 5.1.2 SCALABILITY

Scalability refers to the flexibility to encode one attribute via one latent vector or, alternatively, to decompose one attribute into sub-attributes and encode them by multiple latent vectors. For example, both zoom and translation attributes can be encoded by one latent vector, or they can be decomposed horizontally and vertically and encoded by two latent vectors (see Figure 9).

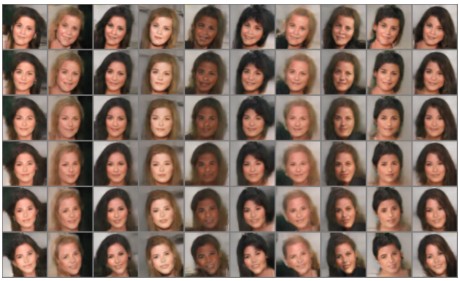

Figure 6: Roll attribute on CelebA dataset.

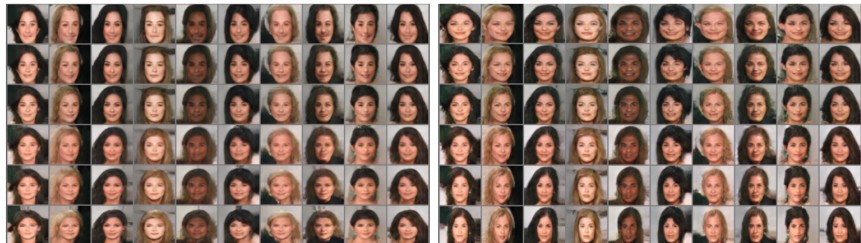

Figure 7: Vertical (left) and horizontal (right) zoom attributes on CelebA dataset.

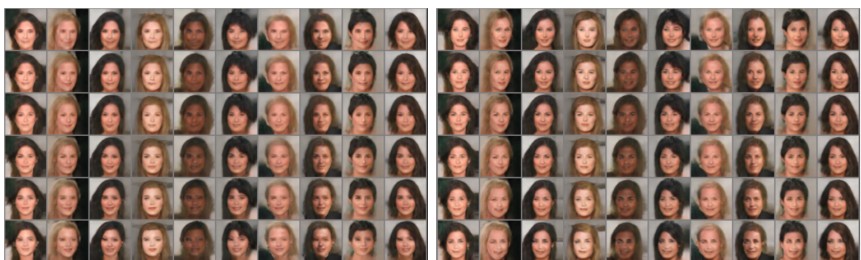

Figure 8: Vertical (left) and horizontal (right) translation attributes on CelebA dataset.

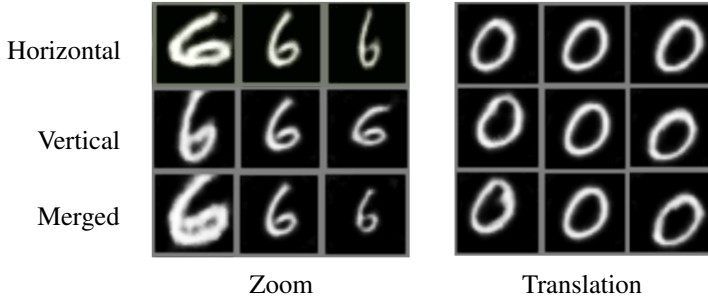

Figure 9: Illustration of scalability. case 1 (row 1 and 2): use two latent vectors to represent horizontal and vertical direction separately. case 2 (row 3): use one latent vector to represent the horizontal and vertical as a merged representation. You may refer to Appendix I for the derivation.

## 5.2 QUANTITATIVE RESULTS

To disentangle object style on dSprites, we choose to model the latent space by one categorical code $c_{\text{cat}} \sim \text{Cat}(K = 3, p = 0.33)$, and 4 continuous codes $c_{\text{cont}} \sim \text{Unif}(-1,1)$ that represent rotation, zoom, horizontal and vertical translation. To the best of our knowledge, ADIS-GAN is the first algorithm that can disentangle shape attributes with categorical codes on dSprites datasets. We have observed that shape disentanglement is easier for larger images. We conducted experiments on the

Table 1: Comparison of the axis-alignment and scalability of the disentangled representation learned by different approaches. ADIS-GAN is the only algorithm that generates disentangled representations with both useful properties.

| Property | $\beta$-VAE | Factor -VAE | $\beta$-TCVAE | Annealed -VAE | Info -GAN | Info -GAN-CR | ADIS -GAN |
|---|---|---|---|---|---|---|---|
| Axis -alignment | no | no | no | no | no | no | yes |
| Scalability | no | no | no | no | no | no | yes |

"Four Shapes" dataset (see Appendix D). Compared to dSprites, the major difference of the "Four Shapes" dataset is that the images are larger. We found that ADIS-GAN can successfully disentangle shape attribute on "Four Shapes" dataset with categorical codes (see Appendix D). The image enlargement procedures and the disentanglement results of dSprites are described in Appendix C.

Table 2: Disentanglement scores on the dSprites dataset. The reference values are from Lin et al. (2019) Table 1. Note that we do not apply model centrality here, which is a hyperparameter tuning method, as our focus is on axis-alignment and scalability. InfoGAN (modified) uses spectrum normalization Miyato et al. (2018) for the discriminator of InfoGAN. For the FactorVAE and BetaVAE scores, we only use continuous codes. For the other scores, we use both continuous and categorical codes. Overall, ADIS-GAN is comparable with the state-of-the-art disentanglement algorithms on the dSprites dataset. We refer to Appendix F for the definations of disentanglement metrics.

| | Model | FactorVAE | DCI | Explicitness | Modularity | MIG | BetaVAE |
|---|---|---|---|---|---|---|---|
| **VAE** | VAE | $0.63 \pm .06$ | $0.30 \pm .10$ | | | $0.10$ | |
| | $\beta$-TCVAE | $0.62 \pm .07$ | $0.29 \pm .10$ | | | $0.45$ | |
| | HFVAE | $0.63 \pm .08$ | $0.39 \pm .16$ | | | | |
| | $\beta$-VAE | $0.63 \pm .10$ | $0.41 \pm .11$ | | $0.21$ | | |
| | ChyVAE | $0.77$ | | | | | |
| | DIP-VAE | | $0.53$ | | | | |
| | FactorVAE | $0.82$ | | | | $0.15$ | |
| | FactorVAE (1.0) | $0.79 \pm .01$ | $0.67 \pm .03$ | $0.78 \pm .01$ | $0.79 \pm .01$ | $0.27 \pm .03$ | $0.79 \pm .02$ |
| | FactorVAE (10.0) | $0.83 \pm .01$ | $0.70 \pm .02$ | $0.79 \pm .00$ | $0.79 \pm .00$ | $0.40 \pm .01$ | $0.83 \pm .01$ |
| | FactorVAE (20.0) | $0.83 \pm .01$ | $0.72 \pm .02$ | $0.79 \pm .00$ | $0.79 \pm .01$ | $0.40 \pm .01$ | $0.85 \pm .00$ |
| | FactorVAE (40.0) | $0.82 \pm .01$ | $\mathbf{0.74 \pm .01}$ | $0.79 \pm .00$ | $0.77 \pm .01$ | $0.43 \pm .01$ | $0.84 \pm .01$ |
| **GAN** | InfoGAN | $0.59 \pm .70$ | $0.41 \pm .05$ | | $0.05$ | | |
| | IB-GAN | $0.80 \pm .07$ | $0.67 \pm .07$ | | | | |
| | InfoGAN(modified) | $0.82 \pm .01$ | $0.60 \pm .02$ | $0.82 \pm .00$ | $0.94 \pm .01$ | $0.22 \pm .01$ | $0.87 \pm .01$ |
| | InfoGAN-CR | $\mathbf{0.88 \pm .01}$ | $0.71 \pm .01$ | $\mathbf{0.85 \pm .00}$ | $\mathbf{0.96 \pm .00}$ | $0.37 \pm .01$ | $\mathbf{0.95 \pm .01}$ |
| | ADIS-GAN | $0.86 \pm .02$ | $0.71 \pm .02$ | $0.78 \pm .00$ | $0.95 \pm .01$ | $\mathbf{0.46 \pm .01}$ | $0.89 \pm .01$ |

## 6 CONCLUSION

This paper introduces the Affine Disentangled GAN (ADIS-GAN) that can explicitly learn the affine transformations via disentangled representations. In contrast to earlier approaches to disentanglement, where inductive biases are not explicit, the disentangled representation obtained by ADIS-GAN is axis-aligned and scalable. The affine regularizer introduced in this paper can be applied to other methods such as the VAE family Kingma & Welling (2013); Higgins et al. (2017); Kim & Mnih (2018); Chen et al. (2018).

Another attractive property of the affine regularizer is that it supports the possibility to utilize expert knowledge as the inductive bias, as it is model based and exploits maximum likelihood estimation. For example, we can disentangle the 2D affine transformations with the decomposed relationships. Task-specific explicit regularizers may provide an alternative pathway for disentanglement compared to existing general implicit regularizers.

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
