# OpenReview forum: "ADIS-GAN: Affine Disentangled GAN"
_ICLR.cc/2021/Conference — Reject_

### Official Review · AnonReviewer4 · 2020-10-29
**Some interesting results, but the benefits of the explicit parameterization are not clear**

**Rating:** 5
**Confidence:** 2

**Review:**

This paper proposes a method that can explicitly learn the affine transformations via disentangled representations in a self-supervised manner. The proposed ADIS-GAN learns to extract affine parameters by adding an affine regularizer on the top of InfoGAN. It seems to me that the main contribution of the paper is to learn the explicit parameterization of the affine transformation, such as rotation, zoom, and translation, by maximum likelihood estimation.  Experiments show that the proposed method is successful in estimating individual parameters of the affine transformation.

**Strengths**
+ The paper represents the first method that can learn explicit parameters of the affine transformation in a self-supervised manner.
+ The experiments show that the proposed method can learn a disentangled representation with explicit affine parameters.

**Weakness**
- Experiments are not convincing enough. The paper only contains experiments on simple/synthetic datasets. The benefits of the proposed method are not completely clear. For the dSprites dataset, the proposed method does not outperform InfoGAN-CR, while InfoGAN-CR can perform disentanglement beyond affine transformations. For scalability, the paper only presents one example for zoom. It is also unclear how to merge zoom parameters into one. It is also helpful to give real applications that can benefit from the explicit disentanglement.
- Writing of the paper can be improved. For example, Section 4.1 needs improvements. The notations are not completely clear, and their roles in the whole algorithm are not clear. It would be better to strengthen the connection of Equation (7) to the affine regularizer loss defined at the end of Section 4.
- The factorization of the affine matrix and derivation of the maximum likelihood estimator are straightforward, making the paper's technical contribution less significant.

**Minor issues**
- The paper emphasizes scalability, but it is not completely clear what scalability the proposed method achieves and how it is achieved. For example, is it possible to merge the translation parameters into a single one? If so, how to do it, and how well the combined parameter performs?
- It is unclear whether the proposed method can factor out skews. Figure 1 shows examples for skews, but the paper does not present how the skew parameters are estimated.
- In Appendix A, the paper assumes the noise statistics for all parameters A_{ij}. It seems unrealistic.
- The citations are correctly formatted. The parentheses are missing for citations.

**Post-rebuttal**

Thank the authors for the rebuttal. It addresses parts of the raised issues. However, my rating keeps the same after reading the rebuttal and other reviews because
1. the contribution and utility of the proposed method are not significant;
2. the writing needs improvement; and
3. the experiments are not convincing enough, and its advantages over previous methods are not clear.

---

> ### Author Response · Authors · 2020-11-20
> **Thank you for your detailed comments, we have added many details in the revised paper (especially Section 4, affine regularizer).**
>
> We would like to thank the reviewer for the detailed and informative comments. We really appreciate this learning opportunity. The responses to the comments are as follows:
>
> A quick summary of the revised paper:
> -	We have revised the whole Section 4, which adds many details to the working principle of the affine regularizer.
> o	We have added Figure 3 and 4 to better illustrate the working principle of the affine regularizer.
> o	We have added a new section 4.1 which describes conversion between latent vectors and affine matrix.
> -	We have added translation examples in Figure 9 to illustrate the scalability property.
> -	We have renamed some of the notations (e.g., W_real to x_real) to make the paper more consistent.
> -	We have added Appendix G that provides information about the network structure.
> -	We have added Appendix H that shows different combinations of affine matrix parameters.
> -	We have added Appendix I to demonstrate how the proposed system achieves scalability.
> -	We have amended some of the citations with \citep{}.
>
>
> Q1: Experiments ... datasets.
>
> Answer: Thank you for your comments. For real-world applicability, perhaps the CelebA dataset can be an indication since it is an RGB dataset with real-world images. We will include more real-world datasets in future work.
>
> Q2: The benefits ... completely clear.
>
> Answer: There are three major contributions:
> -	ADIS-GAN is the first algorithm that can disentangle an entire affine transformation. One of the use cases is the medical image, as discussed in “Explicitly disentangling image content from translation and rotation with spatial-VAE” in Neurips 2019. Their algorithm can only disentangle rotation and translation, where ADIS-GAN can disentangle an entire affine transformation.
> -	Axis-alignment. The latent vectors in previous work do not guarantee the axis-alignment, where the attributes learned may be assigned to different latent vectors for each trial. For instance, if we train on the MNIST dataset with existing algorithms with 3 latent vectors, each trial may assign the rotation attribute to any of the 3 latent vectors. In contrast, in ADIS-GAN we can assign the rotation attribute to the exact latent vector for every trial by predefining how to form the affine matrix using the latent vectors.
> -	Scalability. Previous approaches cannot leverage the compactness and expressivity of the learned attributes. You may refer to Appendix H for more combinations of affine parameters.
>
> Q3: For the ... transformations.
>
> Answer: Thank you for the detailed observations. InfoGAN-CR does not outperform on the MIG score, where our score is 0.46 and InfoGAN-CR is 0.37. For the rest of the score, ADIS-GAN is comparable to the InfoGAN-CR and outperforms other previous approaches. We do not apply model centrality which is a hyperparameter tuning method since our focus is on axis-alignment and scalability.
>
>
> Q4: For scalability ... into one.
>
> Answer: Thank you for the advice. The translation example is added in the revised paper Section 5.1. We have provided more details on how to merge zoom parameters into one in appendix I.
>
> Q5: Writing of ... Section 4.
>
> Answer: Thank you very much for your advice. We have added many details for Section 4 and strengthen the connections.
>
> Q6: The factorization ... significant.
>
> Answer:
> -	Yes, a more complicated assumption might improve the performance and are part of future work (e.g., covariance matrix).
> -	More variations of affine matrix and derivation of the maximum likelihood are added in appendix H and I. We believe the affine matrix family can improve the completeness.
> -	Another technical contribution is the relative affine transformation (see Section 4.2 and Equation 9). We have never seen such a relative transformation in other works.
>
> Q7:  The paper...  single one?
>
> Answer: Yes, it is possible to merge the translation parameters into a single one. We have provided a translation example in the revised paper (Figure 9). You may also refer to Appendix H for more affine matrices combination.
>
> Q8: If so ... performs?
>
> Answer: We have provided a detailed example in Appendix I.
>
> Q9: It is ... estimated.
>
> Answer: Yes, we can factor out skew. The methodology is similar to the way presented in the paper. You may refer to Appendix H.4 to H.7 that includes the skew in the affine matrix.
>
> Q10: In Appendix A ... unrealistic.
>
> Answer: Thank you for your detailed comments. We agree that more complicated assumptions can be made (e.g., the covariance matrix for the noise). Since this is the first work that attempts to solve the affine transformation in a disentangled manner, we would like to make a relatively simpler assumption and verify the idea. We will definitely make more rigorous assumptions in the future.
>
> Q11: The citations ... citations.
> Answer: Thank you for the advice. We have amended it in the revised paper.

---

### Official Review · AnonReviewer1 · 2020-10-29

**Rating:** 4
**Confidence:** 2

**Review:**

Paper summary:

This paper seems to be about aligning latent variables in a GAN with certain affine transformations of the image. This is done by adding an additional "affine regularization" on top of the InfoGAN formulation. This affine regularization seems to be the L2 distance between the randomly sampled c (interpretable latent variables?) and some transformed c'. How the transformed c' is computed was not understood by the reviewer, but it seems to depend on a set of explicitly chosen affine transformations. The affine transformations specified are rotation, horizontal and vertical zoom, horizontal and vertical translation. The authors experiment on MNIST, dSprites, four shapes, and CelebA data. Quantitative metrics are reported for dSprites.

Strong points:

Not many papers tackling disentanglement in GANs.

The paper seems to show nice visual results and seems to be working at recovering the selected affine transformations of images.

Weak points:

(My overall concerns are about the additional supervision required by the method and the clarity of the writing, in particular the method itself.)

The paper mentions this is a self-supervised approach, but it seems closer to a supervised approach where the affine loss can only help recover specific latent factors that must be known beforehand. Would ADIS-GAN be capable of discovering unknown latent factors?

The paper repeatedly claims in the text and in Table 1 that previous works do not have axis-alignment or scalability. Does ADIS-GAN guarantee alignment in some way that is more reliable than in previous works? Isn't the scalability aspect (modeling horizontal and vertical zoom) explicitly encoded by ADIS-GAN, whereas previous works attempt to learn this from data?

I could not follow the exposition in the paper. For instance:
  - There seems to be a complete change in notation from Sections 1-3 to Section 4. No mention of the previously defined x_real, x_fake, z, or c are used in Section 4, where the core contribution is described.
  - Many sentences end in "see Appendix", but I have looked at the Appendix and still come away confused..
  - Why is M 3x3?
  - Where do the A_{ij}'s come from in equation 9? I guessed it might be elements of c, but there are 5 elements in c and 6 A_{ij}'s.
  - There seems to be derivations of a maximum likelihood estimate for constructing c', but these depend on knowing what the A_{ij}'s are.
  - The purpose of Norm_{AL/LA} is never explicitly mentioned, though I imagine it's for scaling the interval [-1, 1] to specific bounds so that M can be applied?
  - What is Encoder in Algorithm 1 and why does it return a linear operator? (Is this different from the encoder in InfoGAN which outputs a distribution over the latent variables?)

Additional feedback:

 - There is mention that ADIS-GAN used categorical codes as well. I assume these do not contribute to the affine loss?

 - There are multiple rows in Table 2 that reference models not explained in the text. e.g. references for HFVAE and ChyVAE, and what is "InfoGAN(modified)"?

 - Please use \citep to include brackets around your citations, when not explicitly using them in a sentence.

---

> ### Author Response · Authors · 2020-11-20
> **Thank you for your detailed comments, we have added many details in the revised paper (especially Section 4, affine regularizer).**
>
> We would like to thank the reviewer for the detailed and informative comments. We really appreciate this learning opportunity. The responses to the comments are as follows:
>
> A quick summary of the revised paper:
> -	We have revised the whole Section 4, which adds many details to the working principle of the affine regularizer.
> o	We have added Figure 3 and 4 to better illustrate the working principle of the affine regularizer.
> o	We have added a new section 4.1 which describes conversion between latent vectors and affine matrix.
> -	We have added translation examples in Figure 9 to illustrate the scalability property.
> -	We have renamed some of the notations (e.g., W_real to x_real) to make the paper more consistent.
> -	We have added Appendix G that provides information about the network structure.
> -	We have added Appendix H that shows different combinations of affine matrix parameters.
> -	We have added Appendix I to demonstrate how the proposed system achieves scalability.
> -	We have amended some of the citations with \citep{}.
>
>
> Q1: The paper ... latent factors?
>
> Answer: ADIS-GAN is a self-supervised method since there are no manual labels required for each image. Yes, the affine transformation matrix is known beforehand, which is similar to many PDE based methods (e.g., physics informed deep learning) where the PDE equations are known beforehand. The ADIS-GAN is also capable of discovering unknown latent factors, where we can define additional latent vectors besides the affine transformation latent factors, those latent factors will learn the unknown latent factors in a completely unsupervised manner just like InfoGAN.
>
> Q2: The paper ... previous works?
>
> Answer: Yes, the latent vectors in previous work do not guarantee the axis-alignment property, which means the attributes learned by the algorithm may be assigned to different latent vectors for each trial. For instance, if we train on the MNIST dataset with existing algorithms and define 3 latent vectors, each trial may assign the rotation attribute to any of the 3 latent vectors. In ADIS-GAN we can assign the rotation attribute to the exact latent vector by predefining how to form the affine matrix using the latent vectors. The rotation attribute will be assigned to the fixed latent vector for every trial.
>
>
> Q3: Isn't the scalability... from data?
>
> Answer: ADIS-GAN provides an inductive bias that makes it possible to learn the data in a scalable way. After training, ADIS-GAN does not rigidly remember the affine transformation itself but the way to conduct the affine transformation and keep the transformed images look realistic as well.
> Previous approaches cannot leverage the compactness and expressivity of the learned attributes explicitly. You may refer to Appendix H and I for how to derive the scalability.
>
> Q4: There seems ... is described.
>
> Answer: Sorry for the inconvenience. We have renamed all the notations that are not consistent in the revised paper. Specifically, all the images are renamed as $x$ instead of $W$. we have also modified the main network diagram (see Figure 2) to make it more comprehensive.
>
> Q5: Why is M 3x3?
>
> Answer: Indeed, if we do not include translation in the affine matrix, it can be 2x2. We have listed more forms of affine matrix in Appendix H.
>
>
> Q6: Where do ...  6 A_{ij}'s.
>
> Answer: Sorry for the confusion, we have it more clear in the revised paper.  A_{ij}’s is the multiplication of the three affine matrices on the right.  If we multiply those three matrices on the right, we can have a 3x3 matrix, the value of the bottom row is constant “0,0,1”, the rest six parameters are A_{ij}’s.
>
> Q7: There seems ... are.
>
> Answer: Yes, we have put more details in the revised paper. You may refer to Section 4.1.2 for the maximum likelihood estimation. The calculation of A_{ij}’s is mentioned in Section 4.1.1. Figure 3 also describes the complete affine regularization.
>
> Q8: The purpose ... applied?
>
> Answer: That is correct. It is described in algorithm 1 “Latent to affine normalization (LA)” and “Affine to latent normalization (AL)”. We have mentioned more details in the revised paper. Norm_LA is mentioned in Section 4.1.1 and Norm_AL is mentioned in Section 4.1.2.
>
> Q9: What is ... latent variables?
>
> Answer: Sorry for the confusion, we have mentioned more details in the revised paper. Encoder in Algorithm 1 does the same thing in InfoGAN, where the input is an image and output is a distribution over the latent variables. You may refer to Figure 3 for the complete process.
>
> Q10: There is ... affine loss?
>
> Answer: Yes, the categorical codes do not contribute to the affine loss.
>
> Q11: There are ..."InfoGAN(modified)"?
>
> Answer: Sorry for the incompleteness. We have included those references in the revised version. HFVAE and ChyVAE in Section 2, and InfoGAN (modified) in Table 2.
>
> Q12: Please  ... a sentence.
>
> Answer: Thank you for the advice. We have corrected those citations in the revised version.

---

### Official Review · AnonReviewer5 · 2020-11-10
**Description of method incomplete**

**Rating:** 3
**Confidence:** 2

**Review:**

This manuscript presents ADIS-GAN (affine disentangled GAN), a method for learning disentangled affine transformations of images such as rotation, zoom, and translation. The authors enforce this disentanglement using a regularizer that penalizes the difference between affine transformations on latent representations. The method is difficult to follow with many critical details left out. Without a complete description, it’s impossible to fully understand and evaluate this work. Some specific comments and questions follow below.
1.	How is the latent vector, C, used? How does it interact with W_basis and W_real? Must it be 5-d to match with rotation, zoom, and translation? How are additional factors of variability encoded?
2.	How is W_basis learned?
3.	What is Norm_AL
4.	What are the encoder/generator/adversary model architectures and how were they trained? For that matter, what is the encoder listed in algorithm 1? I can’t find any reference to it in the text.
5.	How do you ensure that affine transformations of the latent vector actual produce affine transformations of the image? In Figure 5, it looks like the faces aren’t simply being transformed by zoom, because other attributes clear change. In column 5 from the right, the top face has short hair, an exposed ear, and visible teeth. The bottom face, in contrast, has long hair, covered ear, and no visible teeth. This does not appear to be a simple zoom transformation.

Things that would improve my score:
1.	Fully explain the method, neural network architectures, and training parameters.
2.	Explain how affine transformations of the latent vector are forced to produce matching affine transformations of the image.

---

> ### Author Response · Authors · 2020-11-20
> **Thank you for your detailed comments, we have added many details in the revised paper (especially Section 4, affine regularizer).**
>
> We would like to thank the reviewer for the detailed and informative comments. We really appreciate this learning opportunity. The responses to the comments are as follows:
>
> A quick summary of the revised paper:
> -	We have revised the whole Section 4, which adds many details to the working principle of the affine regularizer.
> o	We have added Figure 3 and 4 to better illustrate the working principle of the affine regularizer.
> o	We have added a new section 4.1 which describes conversion between latent vectors and affine matrix.
> -	We have added translation examples in Figure 9 to illustrate the scalability property.
> -	We have renamed some of the notations (e.g., W_real to x_real) to make the paper more consistent.
> -	We have added Appendix G that provides information about the network structure.
> -	We have added Appendix H that shows a different combination of affine matrix parameters.
> -	We have added Appendix I to demonstrate how the proposed system achieves scalability.
> -	We have amended some of the citations with \citep{}.
>
>
> 1.	How is ... variability encoded?
>
> Sub-question a: How is the latent vector, C, used?
>
> Answer: For the latent vector $c$ sampled from a uniform distribution [-1,1], it is used in two ways:
> Usage 1: it is used to generate fake images and calculate the mutual information loss, which is the same as InfoGAN.
> Usage 2: it is used to generate the designed affine transformation matrix $M^\text{real}_\text{transform}$, and calculate the affine loss $L_\text{affine}$. You may refer to Figure 3 of the revised paper
>
>
>
> Sub-question b: How does it interact with W_basis and W_real?
>
> Answer: You may refer to Figure 3 of the revised paper.
> o	We rename $W_\text{basis}$ to $x_\text{basis}$ and $W_\text{real}$ to $x_\text{real}$ for consistency.
> o	We can encode $x_\text{real}$ to $c_\text{real}^\text{basis}$ through the encoder.
> o	$c_\text{real}^\text{basis}$ can be transformed to $M_\text{real}^\text{basis}$ through Flow 1.
> o	By the assumption that every image can be represented as the multiplication between an affine transformation and $x_\text{basis}$, we have $x_\text{real}  = M_\text{real}^\text{basis} \times x_\text{basis}$.
>
>
> Sub-question c: Must it be 5-d to match with rotation, zoom, and translation. How are additional factors of variability encoded?
>
> Answer:  No, we can use different dimensions. You may refer to Appendix H for more details.
>
> 2.	How is $W_\text{basis}$ learned?
>
> Answer: We have renamed $W_\text{basis}$ to $x_\text{basis}$ for consistency in the revised paper. $x_\text{basis}$ is purely learned from data and does not refer to a particular image in the training dataset. For visualization, we can set all the latent vector $c$ to zero and generate $x_\text{basis}$ for each category.
>
> 3.	What is Norm_AL
>
> Answer: We have revised the paper for this part. You may refer to Figure 4 for more details.
>
>
> 4.	What are the encoder ... the text.
>
> Answer: We have revised the paper for this part. You may refer to Figure 2, where E stands for the encoder, G stands for generator and D stands for the discriminator. The encoder in algorithm 1 can be found in both Figure 2 and Figure 3.
>
>
> 5.	How do you ... zoom transformation.
>
> Answer: Thank you for your detailed observation. We have also observed similar phenomena in other algorithms. For example, in InfoGAN, the presence of sunglass changes with hairstyle (Figure 4 in appendix). In beta-VAE and TC-beta-VAE (Figure 5 in appendix), the lighting condition changes with the baldness attribute, and the hairstyle changes with the face width.
> To some degree, it might also help to improve the realism of the images. For example, in Figure 7 left image, where the length of the face is shortened, the algorithm not only compresses the length of the face but knows to add the neck to make the image more natural.
> The affine transformations should be the main attributes that the latent vectors try to encode, but not rigidly perform the affine transformation to the image. It might be a good thing that the disentanglement is correlated to the realism of the generated images.
>
> Things that would improve my score:
>
> 1.	Fully explain ...parameters.
>
> Answer: Thank you very much for the comments. We have amended those issues in the revised paper as follows:
> -	We have explained the method in a more detailed and consistent way (see Section 4 in the revised paper).
> -	We have included a more complete neural network architecture in appendix G.
> -	We have included the training parameters at the beginning of Section 5 (e.g., learning rate, batch, regularization weight).
>
> 2.	Explain how affine ... the image.
>
> Answer: Thank you very much for the comments. We have added a lot of details in the revised paper regarding that. You may refer to Section 4, Figure 3 and 4 for a detailed description.

---

### Decision · Program_Chairs · 2021-01-07
**Final Decision**

**Decision:**

Reject

**Comment:**

Most of the reviewers had serious problems with clarity to start out.
The authors have addressed some, but not all of these problems.

More importantly, there were issues of significance and experimental evaluation.
I concur with r4 on the experimental evaluation.
I think if you're going to explicitly specialize toward disentangling affine transform parameters,
that's fine, but then you're in application-paper land, and I think there needs to be more of an attempt to show
that it will work "in the wild".
For this reason, and for the general reason that reviewers unanimously voted to reject, I am recommending rejection.